

# Predictive nomogram for lymph node metastasis and survival in gastric cancer using contrast-enhanced computed tomography-based radiomics: a retrospective study

Weiteng Zhang[*], Sujun Wang[*], Qiantong Dong, Wenjing Chen, Pengfei Wang, Guanbao Zhu, Xiaolei Chen and Yiqi Cai

Department of Gastrointestinal Surgery, The First Affiliated Hospital of Wenzhou Medical University, Wenzhou, China

[*] These authors contributed equally to this work.

## ABSTRACT

**Background**. Lymph node involvement significantly impacts the survival of gastric cancer patients and is a crucial factor in determining the appropriate treatment. This study aimed to evaluate the potential of enhanced computed tomography (CT)-based radiomics in predicting lymph node metastasis (LNM) and survival in patients with gastric cancer before surgery.

**Methods**. Retrospective analysis of clinical data from 192 patients diagnosed with gastric carcinoma was conducted. The patients were randomly divided into a training cohort ($n = 128$) and a validation cohort ($n = 64$). Radiomic features of CT images were extracted using the Pyradiomics software platform, and distinctive features were further selected using a Lasso Cox regression model. Features significantly associated with LNM were identified through univariate and multivariate analyses and combined with radiomic scores to create a nomogram model for predicting lymph node involvement before surgery. The predictive performance of radiomics features, CT-reported lymph node status, and the nomogram model for LNM were compared in the training and validation cohorts by plotting receiver operating characteristic (ROC) curves. High-risk and low-risk groups were identified in both cohorts based on the cut-off value of 0.582 within the radiomics evaluation scheme, and survival rates were compared.

**Results**. Seven radiomic features were identified and selected, and patients were stratified into high-risk and low-risk groups using a 0.582 cut-off radiomics score. Univariate and multivariate analyses revealed that radiomics features, diabetes mellitus, Nutrition Risk Screening (NRS) 2002 score, and CT-reported lymph node status were significant predictors of LNM in patients with gastric cancer. A predictive nomogram model was developed by combining these predictors with the radiomics score, which accurately predicted LNM in gastric cancer patients before surgery and outperformed other models in terms of accuracy and sensitivity. The AUC values for the training and validation cohorts were 0.82 and 0.722, respectively. The high-risk and low-risk groups in both the training and validation cohorts showed significant differences in survival rates.

Corresponding author
Yiqi Cai, cyqzjwz908123@126.com

**Conclusion**. The radiomics nomogram, based on contrast-enhanced computed tomography (CECT ), is a promising non-invasive tool for preoperatively predicting LNM in gastric cancer patients and postoperative survival.

## BACKGROUND

Digestive cancer continues to be the primary cause of death globally (*Shi et al., 2018*; *Ma et al., 2016*; *Zhang et al., 2021*), with gastric cancer ranking as a prevalent malignancy and the second leading cause of cancer-related mortality worldwide (*Bray et al., 2018*). The presence of peri-gastric lymph node metastasis (LNM) stands as an independent prognostic factor for gastric cancer (*Bando et al., 2002*; *Deng et al., 2014*), emphasizing its critical role in developing standardized and effective treatment approaches for this condition.

Currently, the assessment of lymph node status in gastric cancer patients involves the use of endoscopic ultrasonography (EUS), computed tomography (CT), and magnetic resonance imaging (MRI). However, these imaging modalities exhibit significant variations in sensitivity and specificity, leading to suboptimal rates of LNM detection. While several molecular diagnostic biomarkers for LNM in gastric cancer patients have been identified (*Hiroshi et al., 2009*), their practical application is impeded by factors such as high costs and technical complexities. Presently, CT serves as the primary imaging tool for preoperative lymph node status evaluation in gastric cancer patients. Nevertheless, with a detection accuracy of only 60% (*Kim, Kim & Ha, 2005*; *Giganti et al., 2017*; *Park et al., 2012*), there is a clear need for a more dependable, and precise approach.

Radiomics pertains to the extraction of quantitative features from digital medical images using specialized algorithms aimed at informing clinical decision-making (*Park et al., 2012*; *Kim et al., 2005*). This approach offers a prospective non-invasive method for assessing tumor heterogeneity by integrating numerous imaging features (*Gillies, Kinahan & Hricak, 2016*; *Aerts et al., 2014*; *O'Connor et al., 2017*). Notably, radiomics is increasingly utilized for cancer screening, subtype classification, lymph node metastasis (LNM) detection, survival prognosis, and treatment response evaluation in the pursuit of personalized medicine (*O'Connor et al., 2017*; *Coroller et al., 2015*; *Banerjee et al., 2015*; *Huang et al., 2016*; *Li et al., 2016*; *Jiang et al., 2018*; *Yoon et al., 2016*). While the texture features of CT images have been linked to survival among gastric cancer patients (*Giganti, Tang & Baba, 2019*; *Fu et al., 2015*; *Badgwell et al., 2016*), the predictive features for LNM remain largely unexplored. Rare of the previously reported radiomics models have accurately predicted the presence or absence of LNM in gastric cancer patients based on CT images, and the relevant studies were adapted from MRI-based research methods (*Van et al., 2017*; *Sun et al., 2015*).

The aim of this study was to identify distinct contrast-enhanced CT (CECT) imaging characteristics to preoperatively assess lymph node metastasis (LNM) in individuals with gastric cancer. Subsequently, a radiomics nomogram was developed by integrating imaging features with clinicopathological traits.
## MATERIALS AND METHODS

### Patients

This study involved 192 gastric cancer (GC) patients, comprising 145 males and 47 females, with a mean age of 65.0 ± 10.5 years. The clinical data was obtained from individuals who underwent gastrectomy at the Department of Gastrointestinal Surgery at The First Affiliated Hospital of Wenzhou Medical University between November 2014 and December 2016. The study adhered to the ethical standards of the Declaration of Helsinki and received approval from the First Affiliated Hospital of Wenzhou Medical University (KY2014-R230). Prior to participation, all patients provided written informed consent. Inclusion criteria encompassed a histopathological diagnosis of GC, assessment of LN status in the postoperative pathological report, and the performance of contrast-enhanced abdominal CT 2 weeks preoperatively. Exclusion criteria comprised patients who had received neoadjuvant chemotherapy or radiotherapy before surgery, lacked high-quality contrast-enhanced abdominal CT images due to artifacts, poor expansion, or imaging manifestations, underwent palliative surgery, or had chronic and heterochronic malignant tumors. Baseline clinicopathological data, including demographic details, clinical indicators, and pathological staging data, were sourced from patients' medical records, covering factors such as gender, preoperative hemoglobin concentrations, preoperative serum albumin levels, platelet-lymphocyte ratio (PLR), neutrophil-lymphocyte ratio (NLR), presence or absence of medical conditions (*e.g.*, hypertension, obesity, and diabetes mellitus), Charlson Comorbidity Index (CCI), Nutrition Risk Screening 2002 (NRS-2002) score, CT-reported LN status, tumor size, tumor location, tumor differentiation, histopathological type, pathological tumor-node-metastasis (TNM) stage, levels of carcinoembryonic antigen, and carbohydrate antigen 19-9 levels.

### Image acquisition, tumor segment isolation, and feature extraction

All participants underwent comprehensive abdominal enhanced CT scanning using a 64-slice spiral CT apparatus (Siemens; Munich, Germany), with a delineated slice thickness ranging from 0.75 to 1.25 mm. The portal phase CT scan images were processed using ITK-SNAP software (version 3.8.0; USA; http://www.itksnap.org/) to semi-automatically delineate the tumor-affected region. Two radiologists collaboratively demarcated the tumor area, and their assessment was subsequently authenticated by a third radiologist with similar expertise. The delineated region of interest (ROI) is presented in Figs. 1A1–1A3; 1B1–1B2. The original CT image and the demarcated ROI were stored as medical digital imaging files in the Nearly Raw Raster Data (NRRD) format. For automated feature extraction, Pyradiomics21, a Python programming environment tool (version 3.7.2; https://python.org/), was utilized. Detailed descriptions of the tumor feature extraction, parameter calibration, and Z-score standardization processes can be found in the Supplementary Material for reference and review.

The CT images were processed using ITK-SNAP (version 3.8.0; http://www.itksnap.org/). The delineation of the gastric tumor region was performed by a skilled general surgeon and subsequently assessed and confirmed by a radiologist. An outlined depiction of the patient-specific ROI is shown in Fig. 1 (subsection A1–A3; B1–B3). The original CT images
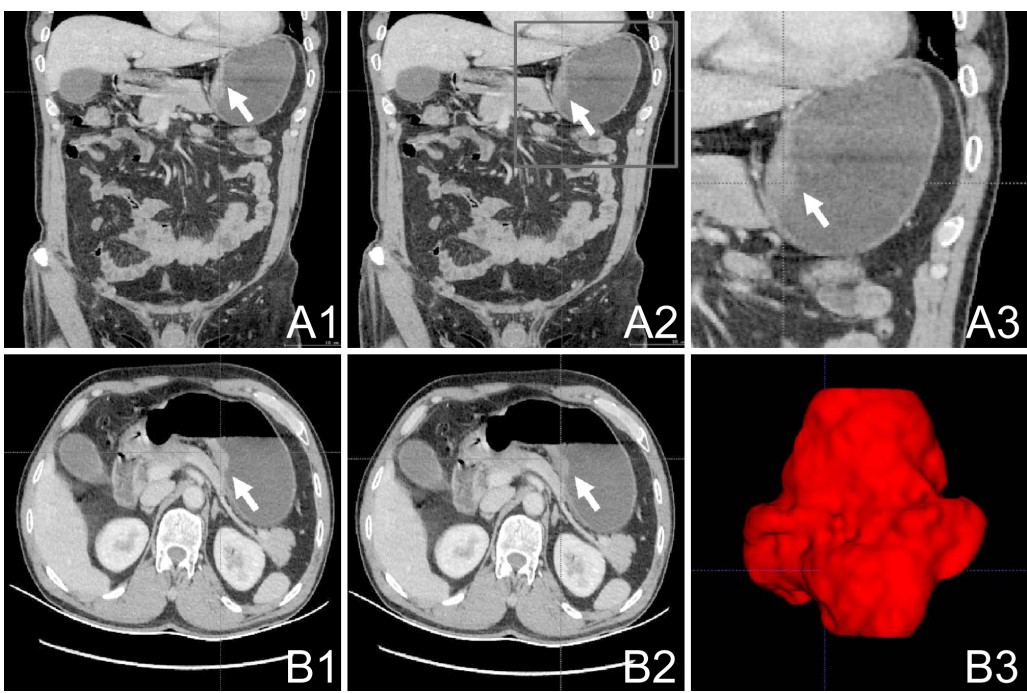

**Figure 1 Work flow of tumor segmentation and feature extraction.** (A1–3): Axial AP enhanced CT showing the tumor (white arrow); (B1–2): Horizontal AP enhanced CT showing the tumor (white arrow); (B3) 3D ITK-SNAP reconstructions of the tumor generated using the 3D Slicer AP, antero-posterior; CT, computed tomography.

and the defined ROI were saved as medical digital imaging files in the NRRD format. Pyradiomics21 in a Python environment (version 3.7.2; available at https://python.org/) was employed for automated feature extraction. Specifics regarding adjustment parameters for feature extraction from the gastric tumor area and the $Z$-score standardization processes can be found in the Supplemental Information.

## Screening of valuable characteristics and establishment of the diagnostic model

The participants were randomly divided into training and validation sets in a 2:1 ratio to ensure robust and generalizable results. This division was performed meticulously and impartially to minimize selection bias, enhancing both the internal and external validity of the study while reducing the risk of systematic differences between the cohorts, thus fortifying the reliability and reproducibility of the results. Random assignment of patients to these sets also strengthened the methodological rigor employed in this research, increasing the credibility of the results.

The outcome variable in the Lasso Cox regression model was the presence or absence of lymph node metastasis (LNM) in gastric cancer patients. The feature selection process employed a robust and systematic approach to identify the most relevant imaging characteristics for the predictive model, including an analysis of all patient cohorts to ensure data validity. Within the training set, feature selection was conducted *via* Lasso-Cox

regression analysis to identify discriminative imaging features. Comprehensive analysis of all 833 characteristics was scrutinized *via* Lasso Cox regression, facilitating the selection of significant features further assessed by logistic regression. Subsequently, the validation subset was used to ascertain the accuracy of the established radiomics-based diagnostic model, addressing concerns about overfitting due to the large number of features in relation to the sample size through cross-validation.

In the cross-validation process, we employed a 10-fold cross-validation technique to assess the robustness and generalizability of the developed radiomics-based nomogram model. Specifically, we utilized a k-value of 10 to partition the dataset into 10 subsets, ensuring that each subset was used as both a training and validation set. By iteratively training the model on k-1 folds and validating on the remaining fold, we obtained an average performance measure, thereby mitigating the impact of overfitting and yielding reliable estimates of the model's predictive capability across different subsets of the data.

To bolster the model's robustness and mitigate potential overfitting pitfalls, and demonstrate a commitment to methodological rigor and data integrity, following the radiomics diagnosis, univariate and multivariate analyses were performed on the diagnostic factors, and receiver operating characteristic (ROC) curves were generated and analyzed to facilitate a comparative assessment of distinct diagnostic models.

## Statistical methods

To ensure the even distribution of continuous parameters, the Kolmogorov–Smirnov test was employed. Normal distribution is presented as means ±standard deviations, whereas non-normally distributed data are represented by medians accompanied by interquartile ranges. Logistic regression analysis to assess the radiomic features was conducted using the "glmnet" package of R software. Intergroup differences in continuous variables were assessed using the Wilcoxon rank sum test, while for categorical variables, the chi-squared test or Fisher's exact test was employed.

A visual nomogram model was constructed using data from univariate analyses *via* multivariate logistic regression analysis. The univariate analysis aimed at identifying individual factors significantly correlated with the presence or absence of LNM in GC patients and factors independently influencing LNM. In contrast, the multivariate analysis determined the combined influence of multiple factors on LNM presence or absence, elucidating the independent and collective impact of various variables and their interplay and interdependencies with LNM. This comprehensive analysis of variable selection for the nomogram entailed rigorous univariate and multivariate analyses to identify the most significant predictors of lymph node metastasis (LNM) in GC patients.

The univariate regression analysis revealed several factors significantly correlating with the presence or absence of LNM, including radiomics evaluation, presence of diabetes mellitus, Nutrition Risk Screening (NRS) 2002 score, preoperative hemoglobin level, platelet-lymphocyte ratio (PLR), neutrophil-lymphocyte ratio (NLR), CT-reported lymph node status, and tumor size ($p < 0.05$). These findings demonstrate that LNM prediction depends on multiple factors, including both radiomic features and clinical parameters. Subsequent multivariate regression analysis further refined the variable selection process,

revealing that radiomics evaluation, NRS-2002 score, CT-reported lymph node status, and diabetes mellitus exhibited a significant correlation with the presence or absence of LNM in GC patients ($p < 0.01$).

The inclusion of these variables for nomogram model construction reflects a robust approach to using radiomic and clinical attributes for predicting LNM, thereby enhancing the model's accuracy for preoperative assessment.

The efficacy and accuracy of the finalized model were assessed using the ROC curve, with statistical significance set at $p < 0.05$. The reported $p$-values in the statistical analyses and evaluations were considered two-tailed. These analytical processes were carried out using R software (version 3.6.0; http://www.R-project.org) and the IBM Statistical Package for the Social Sciences Statistics (version 22.0, IBM Corp., Armonk, NY, USA).

# RESULTS

## Clinical characteristics

The demographic and clinical characteristics of all patients are presented in Table 1. The mean age of the cohort was $65 \pm 10.5$ years, and there was no significant difference between the training cohort ($65.2 \pm 10.4$ years) and the validation cohort ($64.6 \pm 10.7$ years) ($p = 0.716$). Additionally, the two cohorts exhibited similarity in terms of BMI, hemoglobin levels, albumin levels, PLR, NLR, sex distribution, hypertension, diabetes, Charlson Comorbidity Index (CCI), and various pathological and tumor characteristics. These findings indicate that the training and validation cohorts were well-matched, allowing for subsequent analyses and model development.

## Construction and evaluation of radiomics-based nomogram
### Construction of the radiomics model

Seven radiomics features were selected in the training cohort using lasso Cox regression based on lambda.min,, including riginal_shape_Maximum2DDiameterSlice, original_shape_SurfaceVolumeRatio, wavelet.LHL_glcm_ClusterShade, wavelet.LHL_ngtdm_Strength, wavelet.LHH_ngtdm_Strength, wavelet.LLL_firstorder_10Percentile, and wavelet.LLL_firstorder_Median (Figs. 2A–2B). The diagnostic performance of the radiomic features for LNM was evaluated by plotting an ROC curve, and the AUC was 0.76, indicating substantial accuracy. Based on the maximum Youden index of the training cohort, the radiomic score of 0.582 was determined as the cut-off, and the patients were stratified into the high-risk and low-risk categories (Figs. 3A–3B). Univariate regression analysis showed that the radiomics score, radiomics features, diabetes mellitus, NRS-2002 score, preoperative hemoglobin level, PLR, NLR, CT-reported LN status, and tumor size were significantly correlated with LNM ($P<0.05$). Furthermore, the radiomics features, NRS-2002 score, CT-reported LN status, and diabetes mellitus were identified as the independent predictors of LMN as per multivariate regression analysis ($P < 0.01$; Table 2), and were combined into a nomogram model to predict LNM within the abdominal cavity of gastric cancer patients (Fig. 4A).

**Table 1** Comprehensive demographic particulars and clinical attributes of both the training and validation cohorts.

| Variable | Total | Training cohort | Validation cohort | P-value[*] |
|---|---|---|---|---|
| Count | 192 | 128 | 64 | |
| Age, y | 65.0 ± 10.5 | 65.2 ± 10.4 | 64.6 ± 10.7 | 0.716 |
| BMI, kg/m$^2$ | 22.7 ± 3.1 | 22.6 ± 2.9 | 22.9 ± 3.4 | 0.516 |
| Hemoglobin, g/L | 119.1 ± 22.5 | 117.3 ± 23.5 | 122.6 ± 20.0 | 0.123 |
| Albumin, g/L | 37.6 ± 4.4 | 37.4 ± 4.8 | 38.1 ± 3.6 | 0.31 |
| PLR | 169.5 ± 104.5 | 175.5 ± 113.2 | 157.5 ± 83.9 | 0.263 |
| NLR | 2.6 ± 1.7 | 2.7 ± 1.8 | 2.5 ± 1.5 | 0.416 |
| Sex | | | | 0.553 |
| Female | 47 (24.5%) | 33 (25.8%) | 14 (21.9%) | |
| Male | 145 (75.5%) | 95 (74.2%) | 50 (78.1%) | |
| Hypertension | 55 (28.6%) | 40 (31.2%) | 15 (23.4%) | 0.259 |
| Diabetes | 22 (11.5%) | 17 (13.3%) | 5 (7.8%) | 0.262 |
| Charlson Comorbidity Index | | | | 0.26 |
| 0 | 95 (49.5%) | 58 (45.3%) | 37 (57.8%) | |
| 1–2 | 87 (45.3%) | 63 (49.2%) | 24 (37.5%) | |
| 3–6 | 10 (5.2%) | 7 (5.5%) | 3 (4.7%) | |
| NRS-2002 | | | | 0.29 |
| 1–2 | 118 (61.5%) | 75 (58.6%) | 43 (67.2%) | |
| 3–4 | 59 (30.7%) | 44 (34.4%) | 15 (23.4%) | |
| 5–6 | 15 (7.8%) | 9 (7.0%) | 6 (9.4%) | |
| CT-reported LN status | 83 (43.2%) | 50 (39.1%) | 33 (51.6%) | 0.099 |
| Tumor size | | | | 0.915 |
| <3 cm | 68 (35.4%) | 45 (35.2%) | 23 (35.9%) | |
| ≥3 cm | 124 (64.6%) | 83 (64.8%) | 41 (64.1%) | |
| Tumor location | | | | 0.844 |
| Cardia | 29 (15.1%) | 18 (14.1%) | 11 (17.2%) | |
| Body | 44 (22.9%) | 30 (23.4%) | 14 (21.9%) | |
| Antrum | 119 (62.0%) | 80 (62.5%) | 39 (60.9%) | |
| Differentiation | | | | 0.599 |
| Well differentiated | 136 (70.8%) | 93 (72.7%) | 43 (67.2%) | |
| Poorly differentiated | 27 (14.1%) | 18 (14.1%) | 9 (14.1%) | |
| Signet-ring cell | 29 (15.1%) | 17 (13.3%) | 12 (18.8%) | |
| Pathological type | | | | 0.423 |
| Non-ulcer type | 22 (11.5%) | 13 (10.2%) | 9 (14.1%) | |
| Ulcerative type | 170 (88.5%) | 115 (89.8%) | 55 (85.9%) | |
| pT stage | | | | 0.973 |
| I | 33 (17.2%) | 22 (17.2%) | 11 (17.2%) | |
| II | 19 (9.9%) | 13 (10.2%) | 6 (9.4%) | |
| IV | 36 (18.8%) | 25 (19.5%) | 11 (17.2%) | |
| V | 104 (54.2%) | 68 (53.1%) | 36 (56.2%) | |

**Table 1** (*continued*)

| Variable | Total | Training cohort | Validation cohort | P-value* |
|---|---|---|---|---|
| pN stage | | | | 0.708 |
| 0 | 71 (37.0%) | 49 (38.3%) | 22 (34.4%) | |
| 1 | 39 (20.3%) | 28 (21.9%) | 11 (17.2%) | |
| 2 | 35 (18.2%) | 22 (17.2%) | 13 (20.3%) | |
| 3 | 47 (24.5%) | 29 (22.7%) | 18 (28.1%) | |
| pTNM stage | | | | 0.883 |
| I | 46 (24.0%) | 32 (25.0%) | 14 (21.9%) | |
| II | 39 (20.3%) | 26 (20.3%) | 13 (20.3%) | |
| IV | 107 (55.7%) | 70 (54.7%) | 37 (57.8%) | |

**Notes.**

Data shown in the table: mean ± standard deviation; N (%); Wilcoxon rank sum test for continuous variables; chi-squared test for count data.

*$P < 0.05$ was statistically significant.

BMI, body mass index; CT, computed tomography; LN, lymph node; NLR, neutrophil-lymphocyte ratio; NRS-2002, Nutrition Risk Screening 2002; PLR, platelet-lymphocyte ratio.

### *Clinical evaluation of the radiomics-based nomogram*

The AUC values of the nomogram in the training and validation cohorts were 0.82 and 0.722, respectively, while those for the radiomics features were 0.717 and 0.686, and for CT-reported lymph node status were 0.663 and 0.65 (Figs. 4B–4C). Therefore, our radiomics nomogram exhibited good discriminatory performance, comparable to that of radiomics features, and superior relative to conventional CT scans.

### *The radiomics score is associated with the overall survival*

Both cohorts' patients were divided into high-risk and low-risk groups according to the radiomics score, and their overall survival was compared using the Kaplan–Meier method. The lasso Cox regression model indicated a significant association between the radiomics score and overall survival in both the training cohort ($P = 0.001$, hazard ratio (HR) = 3.75, 95% confidence interval (CI) = 1.805–7.791) and the validation cohort ($P = 0.013$, HR = 3.670, 95% CI = 1.225–10.995), thereby allowing for the prediction of survival in gastric cancer patients (Figs. 5A–5B).

## DISCUSSION

We selected seven radiomic features from the abdominal CECT images of gastric cancer patients, including original_shape_Maximum2DDiameterSlice, original_shape_SurfaceVolumeRatio, wavelet.LHL_glcm_ClusterShade, wavelet.LHL_ngtdm_Strength, wavelet.LHH_ngtdm_Strength, wavelet.LLL_firstorder_10Percentile, and wavelet.LLL_firstorder_Median, for the preoperative prediction of LNM. Previous research has illustrated the association of certain radiomic features with tumor heterogeneity, microenvironment characteristics, and treatment response, all crucial to cancer progression and metastasis. While the specific biological significance of these features was not directly addressed in this study, it is pivotal to elucidate their potential biological and pathological relevance concerning LNM in gastric cancer.

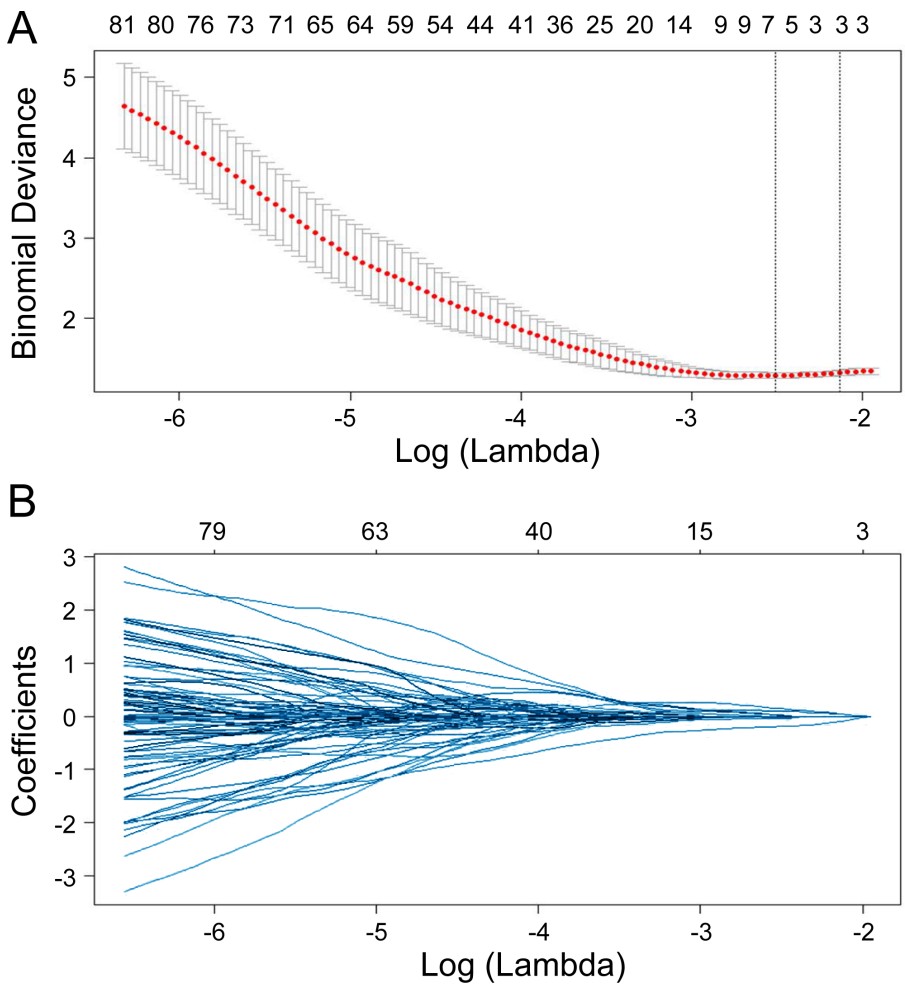

**Figure 2  Lasso regularization plots for variable selection.** The model featuring the most minimal lambda value was chosen (lambda = 0.081, Log (lambda) = −2.507), and seven attributes, identified from the lasso analysis, were integrated into the ensuing logistic regression to formulate the radiomics score.

CT-based radiomics provides a non-invasive and personalized approach for predicting the risk of LNM in gastric cancer patients, distinguishing itself from other diagnostic methods. Despite the well-established prognostic utility of radiomics features, the generalization of these models across different centers is constrained by variations in CT sources and critical characteristics observed in various studies. Our approach involves the development of a universal diagnostic model for LNM using open-source software, contrasting with closed diagnostic systems employed by other researchers. This universal model can be replicated by other centers, making it more suitable for wider hospital populations.

An increasing number of treatment guidelines advocate preoperative neoadjuvant chemotherapies for gastric cancer patients, given that, compared with surgery alone, the addition of neoadjuvant chemotherapy confers a survival advantage without increasing postoperative morbidity and mortality (*Schwarz, 2015*; *Newton et al., 2015*). LNM serves

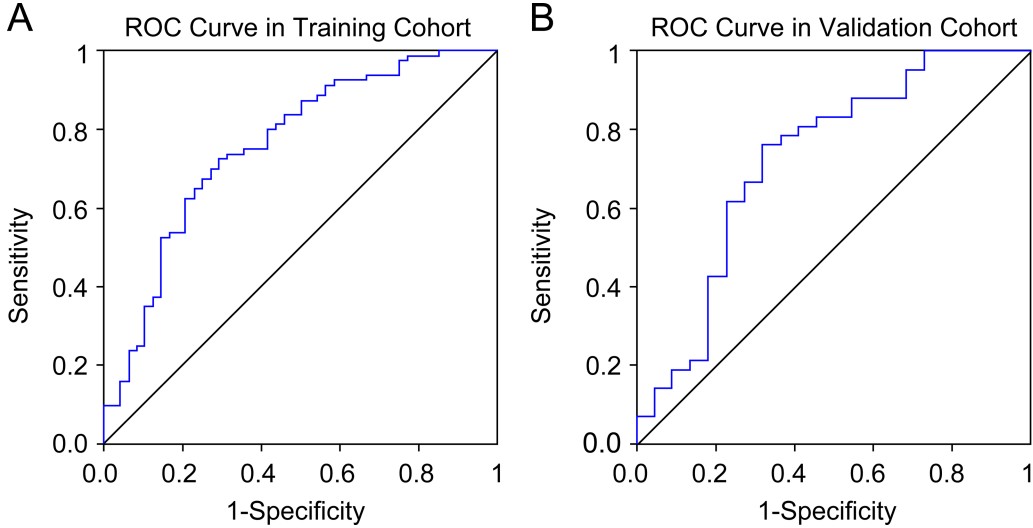

**Figure 3** Receiver operating characteristic curves of the radiomics model in the training and validation cohort.

as a critical determinant of therapeutic interventions for gastric cancer (*Dong et al., 2016*; *Wu et al., 2011*), and preoperative neoadjuvant therapy is routinely recommended for patients with LNM, since it has been demonstrated to downstage the lymph node status and increase the likelihood of achieving R0 resection (*Schuhmacher et al., 2010*; *Mocellin, Marchet & Nitti, 2011*; *Cardoso et al., 2012*). Therefore, the construction of predictive models for the preoperative identification and discrimination of LNM is imperative to establish personalized treatment regimens. Although endoscopic ultrasonography (EUS) is beneficial for local staging of GC, accurately defining the T stage, it demonstrates poor reliability in predicting the presence or absence of LNM (*Philippe et al., 2012*). Likewise, while thin-section CT plays a critical role in preoperative lymph node staging, (*Liu et al., 2020*), the accuracies of both EUS and CT for the preoperative prediction of LNM remain unsatisfactory at 64% and 61–64%, respectively (*Wang et al., 2020*; *Li et al., 2018*). A predictive accuracy of 62% was also computed with routine CT scanning.

The rise of high-throughput data has driven a shift toward precision medicine in cancer diagnostics and treatment, triggering the rapid evolution of alternative approaches such as radiomics, owing to the limitations of traditional imaging techniques. Introduced in 2012, the concept of radiomics has been widely and effectively employed in clinical research (*Fukagawa et al., 2018*). For instance, *Jiang et al. (2018)* was able to identify 15 CT image texture features significantly correlated with preoperative LNM in gastric cancer patients, serving as independent predictors of the pN stage. In our study, we focused on seven imaging features from a total of 833, selected based on their significant correlation with LNM at the smallest lambda value. While our resulting model exhibits greater simplicity, it necessitates further investigation. Additionally, in the MRI domain, Liu et al. analyzed radiomics features and discovered a significant correlation between the whole-lesion

**Table 2  Univariate and multivariate analyses of LNM in patients with gastric cancer.**

| Variable | LN- | LN+ | P-value | |
|---|---|---|---|---|
| | | | Univariate | Multivariate |
| Count | 48 | 80 | | |
| Radiomics score | $-0.2 \pm 1.5$ | $1.0 \pm 0.9$ | <0.001 | |
| Radiomics evaluation | | | <0.001[*] | 0.001[a] |
| High risk | 34 (70.8%) | 22 (27.5%) | | |
| Low risk | 14 (29.2%) | 58 (72.5%) | | |
| BMI, kg/m$^2$ | $23.2 \pm 3.0$ | $22.2 \pm 2.8$ | 0.085 | |
| Age, y | | | 0.138 | |
| <70 | 35 (72.9%) | 48 (60.0%) | | |
| ≥70 | 13 (27.1%) | 32 (40.0%) | | |
| Sex | | | 0.498 | |
| Female | 14 (29.2%) | 19 (23.8%) | | |
| Male | 34 (70.8%) | 61 (76.2%) | | |
| Diabetes | 1 (2.1%) | 16 (20.0%) | 0.004[*] | 0.069[a] |
| Hypertension | 13 (27.1%) | 27 (33.8%) | 0.431 | |
| Charlson Comorbidity Index | | | 0.63 | |
| 0 | 24 (50.0%) | 34 (42.5%) | | |
| 1–2 | 21 (43.8%) | 42 (52.5%) | | |
| 3–6 | 3 (6.2%) | 4 (5.0%) | | |
| NRS-2002 | | | 0.026[*] | 0.049[a] |
| 1–2 | 35 (72.9%) | 40 (50.0%) | | |
| 3–4 | 12 (25.0%) | 32 (40.0%) | | |
| 5–6 | 1 (2.1%) | 8 (10.0%) | | |
| Hemoglobin | | | 0.005[*] | |
| ≥100 g/L | 43 (89.6%) | 54 (67.5%) | | |
| <100 g/L | 5 (10.4%) | 26 (32.5%) | | |
| Preoperative albumin | | | 0.067 | |
| ≥35 g/L | 38 (79.2%) | 51 (63.7%) | | |
| <35 g/L | 10 (20.8%) | 29 (36.2%) | | |
| PLR | | | 0.023[*] | |
| <124.9 | 22 (45.8%) | 21 (26.2%) | | |
| ≥124.9 | 26 (54.2%) | 59 (73.8%) | | |
| NLR | | | 0.026[*] | |
| <2.63 | 39 (81.2%) | 50 (62.5%) | | |
| ≥2.63 | 9 (18.8%) | 30 (37.5%) | | |
| CT-reported LN status | | | <0.001[*] | 0.009[a] |
| Negative | 39 (81.2%) | 39 (48.8%) | | |
| Positive | 9 (18.8%) | 41 (51.2%) | | |
| Tumor size | | | 0.002[*] | |
| <3 cm | 25 (52.1%) | 20 (25.0%) | | |
| ≥3 cm | 23 (47.9%) | 60 (75.0%) | | |

**Table 2** (*continued*)

| Variable | LN- | LN+ | *P*-value | |
|---|---|---|---|---|
| | | | Univariate | Multivariate |
| Tumor location | | | 0.277 | |
|   Cardia | 6 (12.5%) | 12 (15.0%) | | |
|   Body | 15 (31.2%) | 15 (18.8%) | | |
|   Antrum | 27 (56.2%) | 53 (66.2%) | | |
| Differentiation | | | 0.248 | |
|   Well differentiated | 31 (64.6%) | 62 (77.5%) | | |
|   Poorly differentiated | 8 (16.7%) | 10 (12.5%) | | |
|   Signet-ring cell | 9 (18.8%) | 8 (10.0%) | | |
| Pathological type | | | 0.199 | |
|   Non-ulcer type | 7 (14.6%) | 6 (7.5%) | | |
|   Ulcerative type | 41 (85.4%) | 74 (92.5%) | | |

**Notes.**

Data shown in the table: mean ± standard deviation; N (%); Wilcoxon rank sum test and chi-squared test for univariate analysis.

*$P < 0.05$ was statistically significant and included in multivariate logistic regression (forward: wald).

[a]Variables finally in the model (radiomics evaluation, NRS-2002, CT-reported LN status, and diabetes).

BMI, body mass index; CT, computed tomography; LN, lymph node; LNM, lymph node metastasis; NLR, neutrophil-lymphocyte ratio; NRS-2002, Nutrition Risk Screening 2002; PLR, platelet-lymphocyte ratio.

apparent diffusion coefficient histogram and LNM, demonstrating a high predictive accuracy of 82.3% (*Mocellin, Marchet & Nitti, 2011*).

*Wang et al. (2020)* established a CT-based radiomics model for preoperatively predicting LNM in gastric cancer, demonstrating substantial discriminatory power (*Cardoso et al., 2012*). We employed contrast-enhanced CT (CECT) as the foundation for our radiomics models, given its widespread use for preoperative lymph node status evaluation, along with its greater convenience and reliability compared to MRI. CECT showed superior diagnostic ability for LNM compared to routine CT, with AUC values of 0.844 and 0.837 for the training and validation cohorts, respectively. Similarly, radiomics features exhibited enhanced discriminatory ability for LNM compared to routine CT, achieving an accuracy of 80–84%. This aligns with *Wang et al.*'s (*2020*) findings, showcasing the potential of radiomics features to enrich image interpretations and complement routine CT scans for evaluating the lymph node status in gastric cancer patients.

In our study, we integrated the radiomics scores with clinically significant parameters related to LNM, culminating in a nomogram model designed for rapid, convenient, and reliable lymph node analysis to guide personalized treatment. *Wang et al. (2020)* developed a predictive nomogram based on routine CT scans, achieving AUC values of 0.886 in the training cohort and 0.881 in the test cohort, with an 84% accuracy in both (*Cardoso et al., 2012*). Additionally, *Li et al. (2018)* developed a nomogram using intra-tumoral iodine concentration and the Borrmann classification to predict LNM preoperatively, achieving AUC values of 0.76 and 0.793, and corresponding accuracy rates of 0.7 and 0.757 in training and validation cohorts, respectively (*Cardoso et al., 2012*; *Philippe et al., 2012*). Our CECT-based radiomics nomogram model surpassed CECT evaluation and radiomics features in LNM prediction, attaining high AUC values of 0.82 and 0.722

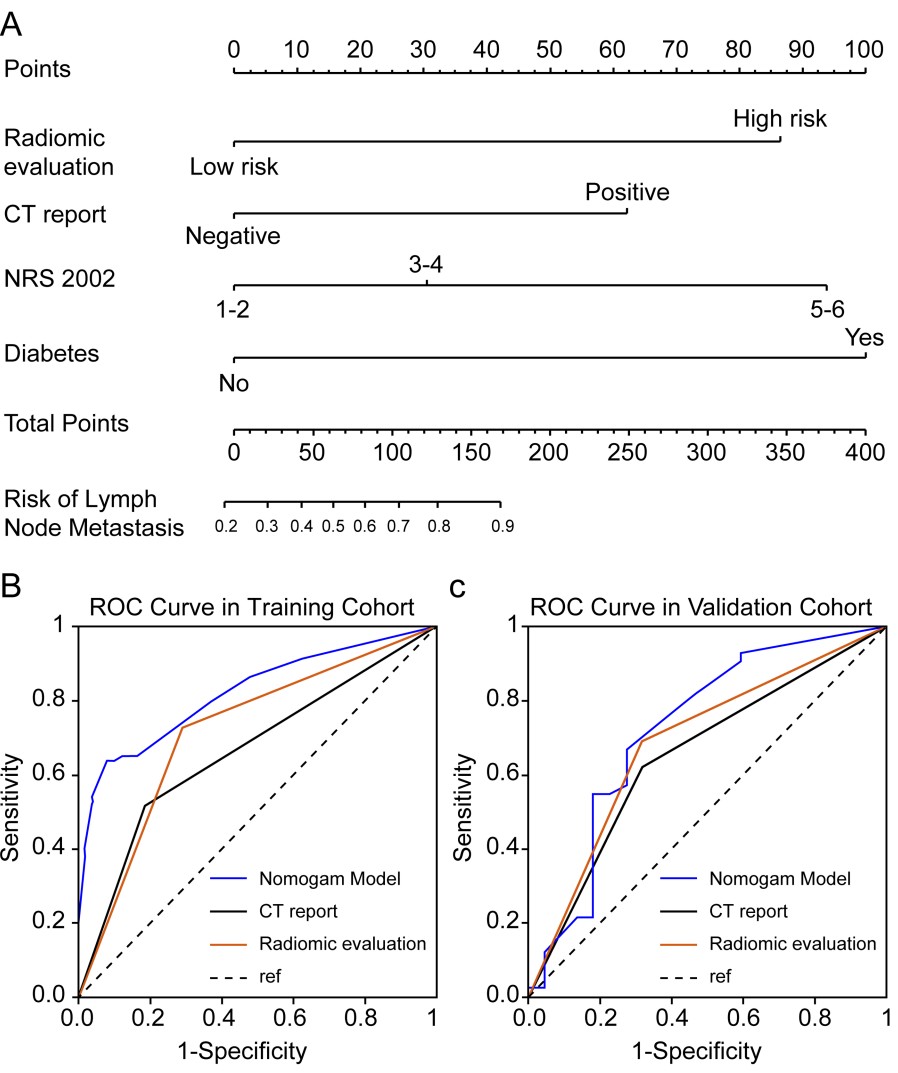

**Figure 4** Enhanced CT-based radiomics nomogram and comparative receiver operating characteristic (ROC) curves. (A) Enhanced computed tomography (CT)-based radiomics nomogram for the prediction of lymph node (LN) metastasis in patients with gastric cancer; (B–C) Comparison of ROC curves of three diagnostic variables (radiomics evaluation, CT-reported LN status, and the nomogram) in the training and validation cohort.

in the training and validation cohorts, respectively. Stratifying patients based on the radiomics scores into high-risk and low-risk groups revealed significant differences in overall survival in both cohorts, highlighting the potential of our novel nomogram model to predict the prognosis of gastric cancer patients. (*Liu et al., 2020*; *Wang et al., 2020*) Radiomics, with its straightforward yet robust visual analysis and routine imaging tools, enables the extraction of parameters that conventional diagnostic methods might overlook. Additionally, a nomogram based on radiomics features can aid clinicians in identifying suitable candidates for neoadjuvant therapy, particularly considering the impact of LNM

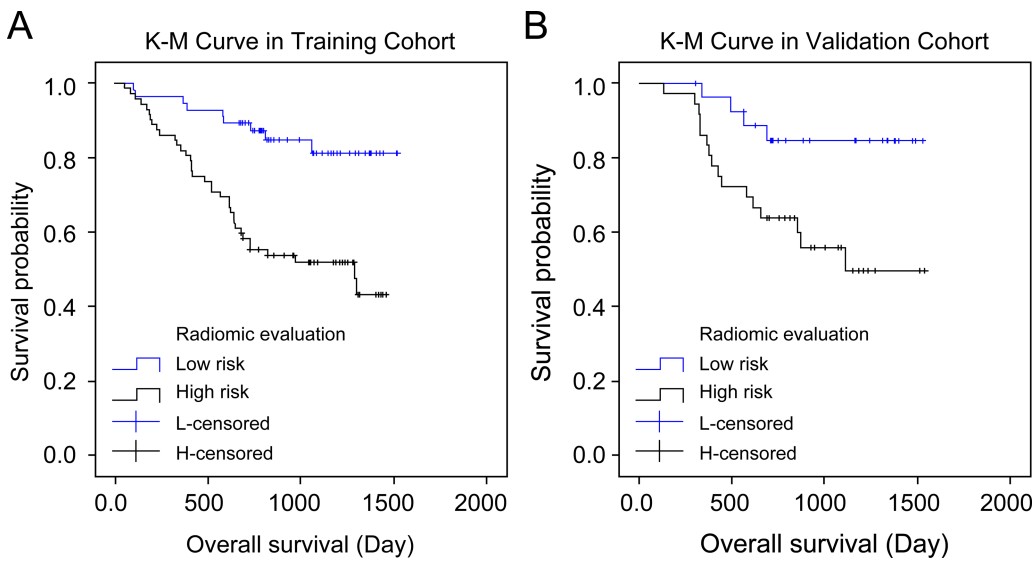

**Figure 5** **Kaplan–Meier curves' overall survival plots.** Comparison of overall survival in the training and validation cohort based on imaging and histology scores: training cohort ($P$ 0.001, hazard ratio (HR) = 3.750, 95% confidence interval (CI) [1.805–7.791]) and validation cohort ($P$ 0.013, HR = 3.670, 95% CI [1.225–10.995]).

on treatment decisions. Overall, our radiomics evaluation system and nomogram model displayed excellent performance in discerning LNM in gastric cancer patients.

This study has several limitations that warrant consideration. Firstly, the retrospective design and single-center cohort constrain the generalizability of our model, necessitating validation in multicenter cohorts. Secondly, our findings were not externally validated, highlighting the need for future studies with external cohorts. Furthermore, we only categorized patients based on LNM status without examining the potential influence of specific stages of metastasis (N1 to N3b) or the anatomical sites of metastasis within the 16 designated locations. Lastly, we acknowledge the inherent limitations associated with the small sample size in the current study, which may impact the generalizability and robustness of the findings. As such, future studies involving larger, multicenter cohorts are essential to validate the predictive performance of the radiomics nomogram model and its utility in clinical practice.

## CONCLUSION

The nomogram model based on radiomics data could be beneficial for preoperative prediction of LNM and postoperative survival analysis of gastric cancer patients.

## ACKNOWLEDGEMENTS

The authors would like to thank Editage for their language editing support.

### Funding

This study was supported by the Department of Health of Zhejiang Province, China (grant numbers Y2100660 and 2016DTA006), the Zhejiang Provincial Health Department Medical Support Discipline - Nutrition (11-ZC24), the Wenzhou Municipal Science and Technology Bureau (Y2020732 and Y20170104). The funders had no role in study design, data collection and analysis, decision to publish, or preparation of the manuscript.

### Grant Disclosures

The following grant information was disclosed by the authors:

The Department of Health of Zhejiang Province, China: Y2100660, 2016DTA006.

The Zhejiang Provincial Health Department Medical Support Discipline—Nutrition: 11-ZC24.

The Wenzhou Municipal Science and Technology Bureau: Y2020732, Y20170104.

### Competing Interests

The authors declare there are no competing interests.

### Author Contributions

- Weiteng Zhang conceived and designed the experiments, authored or reviewed drafts of the article, and approved the final draft.
- Sujun Wang conceived and designed the experiments, authored or reviewed drafts of the article, and approved the final draft.
- Qiantong Dong performed the experiments, prepared figures and/or tables, and approved the final draft.
- Wenjing Chen performed the experiments, prepared figures and/or tables, and approved the final draft.
- Pengfei Wang analyzed the data, prepared figures and/or tables, and approved the final draft.
- Guanbao Zhu analyzed the data, prepared figures and/or tables, and approved the final draft.
- Xiaolei Chen conceived and designed the experiments, analyzed the data, prepared figures and/or tables, and approved the final draft.
- Yiqi Cai conceived and designed the experiments, analyzed the data, authored or reviewed drafts of the article, and approved the final draft.

### Human Ethics

The following information was supplied relating to ethical approvals (i.e., approving body and any reference numbers):

The execution of this study adhered scrupulously to the ethical standards delineated within the Declaration of Helsinki and procured requisite approval from the First Affiliated Hospital of Wenzhou Medical University (KY2014-R230).

## Data Availability

The raw data is available in the Supplementary File.

## Supplemental Information

Supplemental information for this article can be found online at http://dx.doi.org/10.7717/peerj.17111#supplemental-information.

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
