# Peer review of "Predictive nomogram for lymph node metastasis and survival in gastric cancer using contrast-enhanced computed tomography-based radiomics: a retrospective study"

_PeerJ, doi:10.7717/peerj.17111_

## Round 0.1 · original submission · Major Revisions

The manuscript effectively addresses the research problem but requires improvements in certain areas.

Please provide more detailed demographic and clinical information about the study participants.

Please provide a more comprehensive discussion of the univariate and multivariate analyses, particularly regarding variable selection for the nomogram.

Clarifications are needed in several areas:

Random assignment of patients to the training and validation sets (lines 143-144).

Feature selection process and its application to the entire patient cohort or only the training set (lines 144-151).

Concerns about overfitting due to a large number of features compared to the sample size, and the suggestion to consider cross-validation instead of an arbitrary split.

Outcomes of lasso Cox regression and diagnostic models need clarification.

Differences in purposes of univariate and multivariate analyses, and the incorporation of different factors in each model.

Certain content in lines 184-204 should be relocated to the methods section.

Use of appropriate statistical tests for comparison (line 158).

Specification of the type of p-value used (line 163).

**Language Note:** The review process has identified that the English language must be improved. PeerJ can provide language editing services - please contact us at copyediting@peerj.com for pricing (be sure to provide your manuscript number and title). Alternatively, you should make your own arrangements to improve the language quality and provide details in your response letter. – PeerJ Staff

Reviewer 1 ·

Basic reporting

Overall, the manuscript is written in clear language, but the English language should be improved to ensure that an international audience can clearly understand your text. Some examples where the language could be improved. For example, lines 200-204 – the current phrasing makes comprehension difficult. I suggest you have a colleague who is proficient in English and familiar with the subject matter review your manuscript, or contact a professional editing service.

Experimental design

- In lines 143-144, please clarify that patients were randomly assigned to the training and the validation set.
- Lines 144-151 are a bit confusing. Please clarify if you performed feature selection in all the 192 patients or in the training set only. Did you include all 844 features in the lasso regression model? The number of features is much larger than the number of observations, and the number of observations is very small in your study. I think it will introduce an issue of overfitting. In line 209, the AUC in the training set is 0.820, and decreased to 0.722 in the validation set. Is there a possibility of overfitting here? Because of the limited sample size in your study, I think cross-validation validation would be a better fit than the arbitrary split.
- In line 145, please clarify the outcome of the lasso Cox regression mode. In line 148, please clarify the outcome variable of the diagnostic model.
- In lines 148-151, it is unclear the different purposes of univariate and multivariate analyses. Where were the results in Figures B-C from, were they from the univariate model or the multivariate model? What other covariates were included in each model? To build diagnosis models with radiomics evaluation, CT-reported LN status, and the nomogram, did you include all three factors in the same model? If you want to compare the AUC, sensitivity, and specificity of these three methods, shouldn’t you train models for the three factors separately?
- Some of the sentences in lines 184-204 should be moved to the method section.
- In line 158, to compare two groups, I think the Wilcoxon rank sum test is more appropriate than the Kruskal-Wallis test.
- In line 163, please indicate if a two-sided or a one-sided p-value was used.

Validity of the findings

Since the sample size is limited, and models were not trained separately for radiomics evaluation, CT-reported LN status, and the nomogram. I don't think the current conclusions can be derived from your current results.

·

Basic reporting

The manuscript presents a retrospective study on the utilization of contrast-enhanced computed tomography-based radiomics for predicting lymph node metastasis and survival in gastric cancer patients. The study involves the development of a predictive nomogram model and evaluates its efficacy in preoperative prediction of LNM and postoperative survival analysis. While the manuscript effectively addresses the significance of the research problem and provides a comprehensive overview of the study methodology and findings, there are opportunities for improvement.

Experimental design

1)When describing the patient cohorts, consider providing more context on the demographics and clinical characteristics of the study participants, such as age distribution, gender representation, and other relevant variables. This information can help researchers and clinicians understand the study population and assess the generalizability of the findings.
2)Consider providing a more detailed discussion of the univariate and multivariate analyses, specifically regarding the selection and interpretation of variables included in the predictive nomogram model. This can help readers discern the rationale behind the inclusion of specific clinical and radiomic features in the predictive model.

Validity of the findings

The section on radiomics model building and evaluation provides important insights, but some findings could be better articulated for clarity. For instance, the rationale for the selection of specific radiomic features and the clinical significance of their association with lymph node metastasis could be more explicitly discussed to help readers understand the relevance of these findings.

Additional comments

In table 1and table 2, the yes or no of hypertension and diabetes is actually repeated, you can just put ‘yes’ percentage, so as the positive LN status.

---

## Round 0.2 · Minor Revisions

Thank you for submitting your manuscript to our journal. After careful consideration, it appears that your study offers valuable insights into the utilization of radiomics for predicting lymph node metastasis and survival in gastric cancer. However, before a decision can be made regarding publication, it is necessary to address comments from two reviewers.

1. Please specify what the outcome variable is in the lasso Cox regression model as mentioned in Section 1.2.3.

2. Provide a detailed description of how cross-validation was conducted within your study to strengthen the experimental design section.

3. Address the limitations associated with your study's small sample size and discuss how these may affect the validity of your findings.

4. Address the question raised about the reproducibility of your diagnostic approach, given that your research is single-centered.

5. Make language and terminology adjustments according to reviewer 2 's comments.

Reviewer 1 ·

Basic reporting

- In 1. 2.3 Screening of valuable characteristics and establishment of the diagnostic model, please indicate what the outcome variable is in the lasso cox regression model.

Experimental design

- It's unclear how the cross-validation was conducted. More details descriptions of methods and results are needed.

Validity of the findings

- Given the inherent limitations of a small sample size. It's hard to validate the current study findings.

·

Basic reporting

1. In line 23, 'based' is better than the word 'centric' for the consistency with your title.
2. In the abstract, it is better to add your specific results, for example, your training or validation cohorts' AUC value.
3.In line 46, the word 'analyzing' is a bit redundant, you can consider to delete it.
4.In line 72-73, your statement is too absolute, and there are indeed literature that has used CT imaging radiomics related articles, such as Gao X, Ma T, Cui J, et al. A radiomics-based model for prediction of lymph node metastasis in gastric cancer. Eur J Radiol. 2020;129:109069. doi:10.1016/j.ejrad.2020.109069.

Experimental design

no comment

Validity of the findings

In line 64, you mentioned 'reproducible diagnostic approach'. How can single-centered research demonstrate model repeatability?

Additional comments

no comment

---

## Round 0.3 · accepted · Accept

Since the authors have addressed all concerns, this paper can be accepted for publication.

·

Basic reporting

I think this article basically meets the standards for publication, and I have no other issues to raise.

Experimental design

no comment

Validity of the findings

no comment